# Building Community Resiliency through Immersive Communal Extended Reality (CXR)

**Sharon Yavo-Ayalon** [1,2,*], **Swapna Joshi** [3], **Yuzhen (Adam) Zhang** [2], **Ruixiang (Albert) Han** [4], **Narges Mahyar** [3] and **Wendy Ju** [2]

1. Graduate School of Architecture, Planning and Preservation, Columbia University, New York, NY 10044, USA
2. Jacobs Technion-Cornell Institute at Cornell Tech, New York, NY 10044, USA
3. Manning College of Information & Computer Sciences, University of Massachusetts, Amherst, MA 01002, USA
4. New York University, New York, NY 10016, USA
* Correspondence: sma2265@columbia.edu

**Abstract:** Situated and shared experiences can motivate community members to plan shared action, promoting community engagement. We deployed and evaluated a communal extended-reality (CXR) bus tour that depicts the possible impacts of flooding and climate change. This paper describes the results of seven community engagement sessions with a total of $N = 74$ members of the Roosevelt Island community. We conducted pre- and post-bus tour focus groups to understand how the tour affected these community members' awareness and motivation to take action. We found that the unique qualities of immersive, situated, and geo-located virtual reality (VR) on a bus made climate change feel real, brought the consequences of climate change closer to home, and highlighted existing community resources to address the issue. Our results showed that the CXR experience helped to simulate a physical emergency state, which empowered the community to translate feelings of hopelessness into creative and actionable ideas. Our finding exemplifies that geo-located VR on a bus can be a powerful tool to motivate innovations and collective action. Our work is a first-of-its-kind empirical contribution showing that CXR experiences can inspire action. It offers a proof-of-concept of a large-scale community engagement process featuring simulated communal experiences, leading to creative ideas for a bottom-up community resiliency plan.

**Keywords:** urban interfaces; communal extended reality (CXR); climate change; coastal communities; community engagement; social awareness; environmental awareness; multi-user XR; VR in the car; geo-located VR; digital twin; unity; design; community resiliency plan

## 1. Introduction

Addressing climate change challenges requires collaboration, communication, and the agreement of different stakeholders [1–3]. Even in at-risk coastal communities, which are expected to be heavily impacted by rising sea levels and flooding [4–6], it is difficult for community members to take action because they fail to grasp the impact this change will have on their daily lives [7–9]. Previous research showed that by delivering information and increasing understanding of the causes and consequences of climate change, science could help make climate change feel real [8,10,11]. However, based on psychological research and global politics, sustainability leaders have concluded that science and science communication are not enough to stimulate behavior change or substantive action on the climate crisis [12]. The automated city might take advantage of urban data, digital models, and mixed-reality interfaces to influence community awareness and motivation to address the climate crisis; innovative methods, such as virtual reality (VR) or augmented reality (AR) [9,13,14], may be useful in engaging people on complex social and environmental issues by making climate change more palpable in ways that were not possible 33 years ago [10].

To address this challenge, we deployed and evaluated a communal extended reality (CXR) system to provide immersive experiences that help people envision the likely impacts of floods and motivate them to take action (Figure 1). We invited community members (*N* = 74) to share a ride on the local shuttle bus, taking advantage of the public transport system as a way to bring people together in a familiar environment. They used VR head-mounted displays (HMDs) to glimpse into their neighborhood's past and future, traversing the island's physical space to understand climate change through time. Our CXR was built by utilizing a digital twin (DT) environment of Roosevelt Island, developed earlier in this research project. It used advances in modeling that are critical to associating the visualized depictions with the real-life location. We conducted pre- and post-ride focus groups and surveys to understand how the CXR experience affects community awareness, reveals community perceptions, knowledge, and willingness, and incites motivation to take action. More broadly, we were interested in learning how our CXR experience can be used to inspire community resiliency toward climate change. In this paper, we present the participant's responses to and lessons learned from the inaugural deployment of the CXR (ours, forth coming).

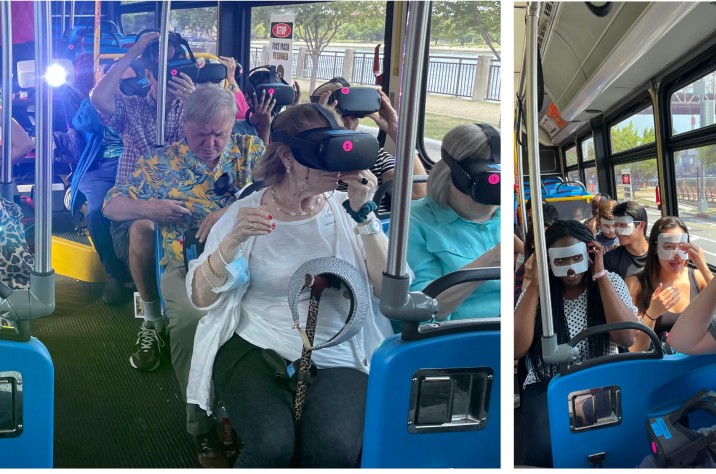
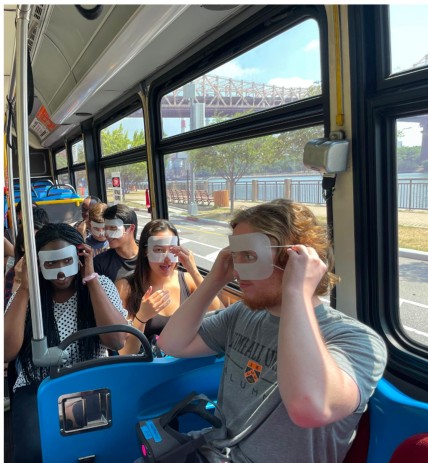

**Figure 1.** Our communal extended reality system enabled community members to experience a "drive-through" simulation of the possible future floods and sea-level rise that will occur due to climate change. This shared experience was used as part of a community engagement process to elicit concerns and plans for neighborhood resiliency planning.

Our research main contributions are three-fold:

1.  An empirical contribution of the results from the inaugural real-world deployment of the CXR system with participants from the community.
2.  A methodological contribution of a community engagement protocol. Our protocol suggests using a mass transport vehicle to simulate climate change scenarios. It included research team engagement with government officials to develop the system and design the experience and bring social groups and individuals from the community during the deployment process.
3.  A practical contribution of a draft of a community resiliency plan based on the creative ideas and strategies discussed by the community.

We begin our paper by reviewing related works about community engagement and climate change and the use of virtual reality for community engagement, which we drew upon in this project. We also discuss the gaps and challenges in community engagement work with VR that we address with this project. We then elaborate on our community engagement protocol and the mixed-methods research design that combines focus groups and surveys with interactive design research. Section 6 presents our analysis of three types of data that we collected, including the focus group transcripts, the bus ride videos, and the survey's quantitative findings. Finally, we discuss our results and limitations compared

to the state-of-the-art community engagement processes toward climate change resiliency. Our conclusions suggest practical takeaways for further developing and deploying CXR experiences in these contexts and opportunities for future improvements. We compiled the participant's suggestions into a draft community resiliency plan (added as an  Appendix A). This draft will be disseminated among the participants and local authorities to initiate steps toward a formal community resiliency plan.

## 2. Related Work

### 2.1. Community Engagement and Climate Change

Community engagement is paramount for driving communities toward a proactive response to climate change [2] and coastal environmental issues [15]. Community engagement activities can help community members, organizations, and governments to understand how climate change affects them, and, in turn, enable these diverse stakeholders to collaboratively develop strategies that are more likely to be accepted and efficacious [16].

Many climate-change initiatives in coastal communities exemplify the benefits of community engagement over top-down planning approaches:

- They promote education and awareness about climate change and the actions that can be taken to address it. Research has shown that participation enables communities to achieve their collective goals, develop knowledge and potential, and make informed and responsible decisions [1,17].
- They bring stakeholders together to develop locally appropriate solutions tailored to that community's specific needs and resources [18]. For example, Harvey et al. [19] describe how the Australian Coastcare program brought together local stakeholders in the decision making and management of coastal resources.
- They can help to identify local priorities for specific climate-change-related issues. Barron et al. [20] and Stevens et al. [21] describe using community engagement to gather input and feedback on proposed adaptation plans to prepare for the flooding and erosion caused by sea level rise in Vancouver, Canada and New South Wales, respectively.
- They enhance ownership and accountability of the initiatives. Krasny and DuBois [22] describe environmental education programs in post-Hurricane-Sandy New York City with planting events and educational tours about green infrastructure, such as wetlands, dunes, and oyster reefs.
- They can rally community members to aid in data collection and analysis. Community-based monitoring programs can be used to gather data on coastal hazards and impacts, such as sea level rise and beach erosion [23,24], to reduce the impacts of coastal hazards and improve water quality [22].

In a few cases, community participation has been found to have hindered climate change initiatives. For example, communities may oppose initiatives that they perceive to be a threat to their property values or quality of life [25,26]. Trust issues between the community and the government [27] may undermine people's engagement with policies, ultimately leading to time delays, resource overuse, and presenting challenges in the implementation of sustainability projects [18]. For coastal communities specifically, a limited understanding of environmental sciences can prevent these communities from comprehending the expected impacts of climate change on their immediate surroundings [8,9]. Despite physical closeness and direct contact with the ocean, some coastal communities still find themselves psychologically distant from the slow processes of climate change [7,28,29]. This can make these communities less concerned about climate-related decision making [30].

For these reasons, in their book, Moser and Dilling [12] argue that traditional science communication may not be sufficient to stimulate proactive behavior toward mitigating the climate crisis. They relate how challenges, such as lack of immediacy, remoteness of impacts, skepticism about solutions, and threats to self-interest, have left people with a sense that climate change is a problem resistant to solution. Sheppard [31] suggests taking newer approaches, such as experiential learning, encouraging place attachment, and

social pressure at the local level, to mobilize community awareness and action on climate change. They suggest using compelling current and future visualizations of climate change to engage communities in planning and help them to self-educate and mobilize for local deliberation and action. Scurati et al. [32] emphasize VR to encourage public participation, and allow stakeholders, such as planners and community members in performing a real-time analysis of environmental issues and the social impacts of their actions in the early stages of smart city planning.

### 2.2. VR for Climate Change Community Engagement

There have been some attempts to use VR simulations to help communities visualize the potential impacts of climate change. Many of these systems have been deployed as VR walkthroughs [33] to showcase proposed renewable energy projects, such as wind or solar installations, to address community concerns. Other deployments have been in the form of shared VR experience systems developed [34] at room scale for multiple people to experience in a physically co-located studio setting and immersive apps and games [35] that provide virtual overlays of proposed coastal protection measures, such as dunes or seawalls, on real-world images to gather feedback on the design. Finally, as relevant to our work, VR tours or field trips [10] have been used to showcase successful climate change initiatives adopted by other communities to inspire people to adopt these initiatives in their neighborhoods.

VR has supported community engagement on climate change by (1) delivering information and increasing understanding of the causes and consequences of climate change [10,11], (2) allowing communities to feel emotionally connected by sharing their first-hand or past experiences [36,37], and (3) providing opportunities for them to participate, act, mitigate and adapt [38]. Studies use VR imaging to show, for example, the consequences of ocean acidification [39], or environmental damage from water consumption [36], deforestation [40], and plastic pollution [41], suggesting how physical and visual exploration in VR can lead to a greater understanding of climate change and the risks associated with it.

Research has also shown that VR immersive qualities achieved by photogrammetry, artistic work, and three-dimensional models increased proactive behaviors [42,43]. For example, Jude et al. [43] found that when locations portrayed in VR were familiar to users, it increased stakeholders' awareness, initiated dialogue on more immediate issues for the community, and triggered stronger emotional reactions. Calil et al. [42], based on a study of three coastal communities' use of shared VR for communication and community outreach, found that using VR significantly increased engagement with planning and conversations when compared with traditional processes. Moreover, VR made the experience realistic [42], entertaining and impactful [11,14,42]. Shared VR experiences with multiple stakeholders encourage audiences to concentrate on their common interests and have been helpful for environmental fundraising [42,44].

### 2.3. Challenges in Using VR for Community Engagement

Prior work by Marques et al. [45] Speicher et al. [46] and Sereno et al. [47] highlighted the significance of considering aspects of collaborative work in the use of mixed reality, such as the location and positioning of members, expertise and role of interactors within the group, amount of time used in visualizing the mixed reality, symmetry [48], technology use and interaction effort techniques, and the context of use. Sereno et al. [47] emphasize that social presence, i.e., being with others and engagement though interaction with the same data, influences the collaborative outcome of the systems. They highlight how feelings of transportation are difficult to achieve in co-located systems.

However, there are additional social and technical considerations to using VR at the community level. First, visualizing adaptation that appropriately balances realistic options and visionary concepts have been challenging, as stakeholders tend to disagree on the details of the visuals or the narration accompanying them [42,43]. Additionally,

VR experiences have thus far been projected asynchronously with stakeholders [1,42], not allowing them to share experiences [49] or have co-located interactions.

Existing examples of city-scale DTs [50–52] include models with buildings, infrastructure, vegetation, terrain, and other elements. Some of them offer ways to simulate information from various processes onto them, such as those used in the planning of Virtual Singapore, SideWalks' efforts in Toronto, CityZenith planning of Amaravati [53], Glasgow's Future City initiative or Cambridge University's National DT project. Lehtola et al. [54] suggest that DTs for cities help authorities to understand the organization of everyday services in the city to perform planning activities, for example, to increase productivity in city services and to facilitate response to crises, such as climate change in Helsinki. However, Shahat et al. [55] suggest that the complexity of the city and the relationship between its elements, such as humans, infrastructure, services, and technologies, have made it a challenge to use these DTs for communication with communities, Platforms using DTs to enable community engagement are still being envisioned [56].

Ens et al. [57], in their review of collaborative mixed-reality systems over time, point to the need for research to take new directions, including the need for using these systems for real-world social groups [58] and considering how their social roles may change their experience and interaction with technology. They also recommend using transitional interfaces that overlay physical environments and artifacts to encourage conversations and feelings of connection between interactors.

While the use of the VR experience for engaging the community in learning, understanding, and responding to climate change, in coastal areas and in locations familiar to stakeholders, has been attempted in previous studies, most studies were implemented in indoor, lab-like environments, with individuals or groups of participants called in for experiments or interviews. In the following section, we describe the methods that allow us to evaluate many understudied, situated, communal and immersive dimensions for a mixed-reality system.

## 3. Community Engagement Process and Timeline

### 3.1. Case Study on Roosevelt Island Community

This community engagement project took place on Roosevelt Island, in New York City. Roosevelt Island is an apt site for community engagement activity; historically, the island is a site of utopian urban planning, planned by Johnson and Burgee in 1969 to integrate people of different income levels and age groups [59–61].

One iconic aspect of the local island culture which emerged from Johnson and Burgee's innovative master plan was the Roosevelt Island Red Bus Shuttle, which provides residents with free transportation 24 h a day. The bus is wheelchair-accessible, and traverses nearly the entire length of the island, from the Coler long-term chronic care hospital in the north to the Franklin Delano Roosevelt Four Freedoms Park in the south.

Roosevelt Island faces imminent danger of flooding from climate change; the NYC Department of City Planning Projections shows that a 100-year flood will submerge most of Roosevelt Island (Figure 2). As unnerving as this map is, the disconnect between the real risk and people's everyday concerns prevents it from feeling urgent. It does, however, have implications for emergency preparations, individual and municipal decision making, and community resiliency.

The residential community on the island maintains its local identity and unique traditions. The residential community on the island has highly involved activist groups that monitor and voice opposition to the state-led operating corporation decisions [60], advocate for the accessibility needs of the island's diverse population of people with disabilities (Roosevelt Island Disabled Association homepage, 2023. https://www.ridainc.org/home (accessed on 2 March 2023)), and champion protections for local wildlife (Wildlife Freedom Foundation homepage, 2023 https://www.wildlifefreedomfoundation.org (accessed on 2 March 2023)). These efforts demonstrate the community's commitment to environmental protection and their potential to make a positive impact to mitigate climate change.

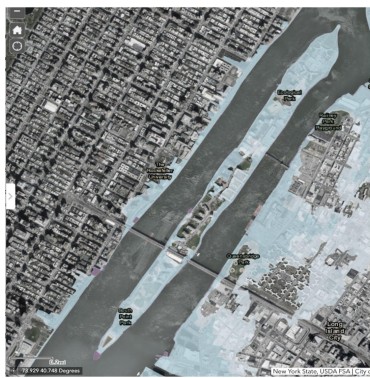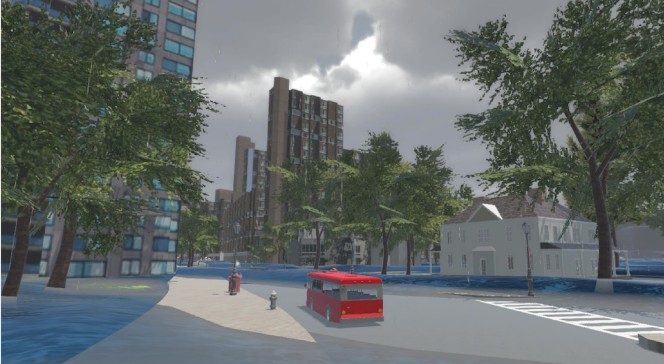

**Figure 2.** Roosevelt Island in NYC flood map and the view from our digital twin.

### 3.2. Focus Group Activity Design

We engaged for eight months with different community stakeholders and used their feedback and insights to design our CXR experience (ours, forth coming). We designed the system through a series of meetings with a team of officials composed of the manager of the operating corporation, the chief of staff, the director of communications and community affairs, the director of parks and recreation, and the director of transportation. They advised our multi-disciplinary research team on the use of the bus and provided feedback on our conceptualization of the CXR system (Table 1).

**Table 1.** Community engagement process and timeline.

| Dates | Phase | Activity | Outcomes |
|---|---|---|---|
| November 2021–January 2022 | Preliminary Research | Meetings with climate change experts, urban designers, and city officials to learn about the needs and existing tools. | Executive summary with main objectives. Draft script for the immersive experience. |
| February 2022–May 2022 | Development | Developing the immersive experience using Unity. Running test drives to check the efficacy of the experience. Developing the questioners of before and after the experience. | A prototype of the immersive experience. |
| June 2022–August 2022 | Experiment | Running 7 RIOC shuttle bus ride with 15 people in each, followed by focus group. | Data sets composed of videos |
| September 2022–October 2022 | Analysis | Analyze data and footage captured from the experiment. Observing changes in point of view and understanding possible causal factors. | Policy report Immersive exercise for RIOC to share with the residents. |

### 3.3. Experience Design

During our community engagement process, we became aware of some concerns that the realistic visualization provided by our CXR system might "scare people off living on the island". We faced similar pushback from members of the climate change impact groups who refused to participate, saying that such a "doomsday experience" depresses people and does not encourage them to be proactive.

Another climate activist cautioned us about showing any fearful end-of-day climate scenarios and suggested using more positive scenarios of alternatives for action. Considering this feedback, we carefully curated the abundance of city and state open-source data to create a 15 min script anchored on scientific forecasts (Figure 3). We emphasized several times throughout the ride that we present speculated effects and reminded participants that we can and should act today to mitigate this speculated future.

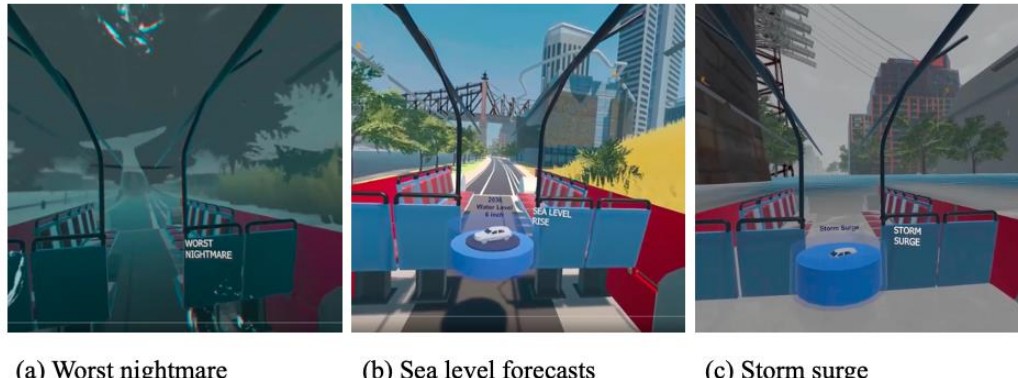

(a) Worst nightmare     (b) Sea level forecasts     (c) Storm surge

**Figure 3.** Top-rated scenarios, based on survey of the participants.

To bridge the divide between researchers and participants, we included two narrators on the bus—a researcher who acted as the tour guide, in conversation with the "expert", a pre-recorded voice-over that shared scientific information. A draft of the elaborated script was sent to three stakeholders from the community, including a climate change expert and other community members who were not on our participant list. They added their thoughts and comments, and we revised the script accordingly.

We highlighted in our script that even though the forecasted floods and storm surges may not happen in the participant's lifetime, they could act to prevent them. A significant part of the tour was dedicated to flood scenarios to allow participants to learn some terms related to floods and how to act if they were ever in this situation.

## 4. Method

### 4.1. Three-Part Group Activity

To understand how shared communal experience influences community resilience planning, we developed a three-part experience (Figure 4):

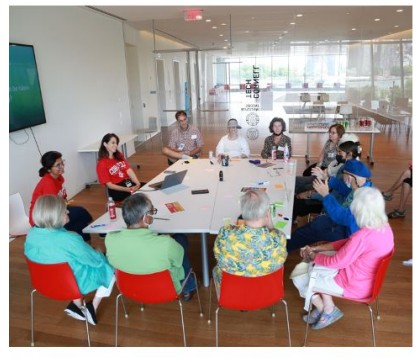
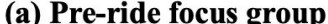
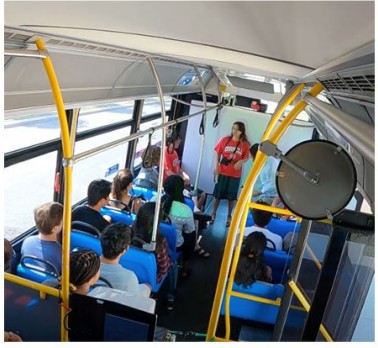
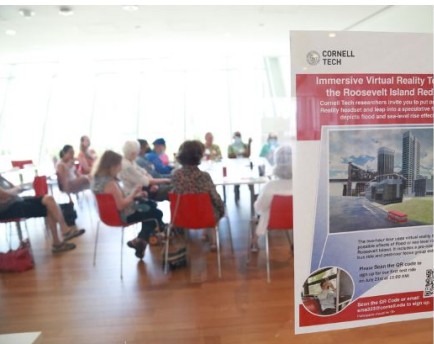

**(a) Pre-ride focus group**     **(b)The Immersive Bus Ride**     **(c) Post-ride focus group**

**Figure 4.** Assessing the effect of providing shared communal experiences with pre- and post-ride focus groups.

1. Pre-ride focus group and survey: A 30 min session on understanding the participant's knowledge and thoughts about climate change and flood risks on RI before the ride.
2. The immersive CXR bus ride: A 30 min ride to complete a full circle around the island—half of the ride conducted through the geo-aligned cinematic movie, while participants wear HMDs VR, with the second half returning to the starting point, while the participants rest and contemplate.
3. Post-ride focus group and survey: A 30 min session aimed to learn if and how the experience changed participants' knowledge, thoughts, and understanding of climate

change and flood risks and to collect their ideas on practical actions toward developing a community resiliency plan.

All aspects of the study were approved by our institution's internal review board (IRB).

### 4.2. The CXR System

The communal extended reality (CXR) system features (1) VR head-mounted display (HMD) for the participants, (2) a fallback system that includes a projector and a screen, for those who do not wish to use the HMDs, and (3) a 3D model "digital twin" of the environment. The system synchronizes the multi-user VR HMDs and provides a geo-located, situated, shared experience for people who are riding together on a moving platform.

In the CXR system's inaugural run, participants riding together on RI Red Shuttle's normal route experience a 360-degree panoramic video of the island's digital twin, which is synchronized to the actual location of the bus. This video features the island under different climate change and flooding conditions. The experience was designed to help people envision the likely impacts of floods by glimpsing into their neighborhood's past and future to understand climate change through time as they traversed the island's physical space. On the ride, the participants hear an informational soundtrack, which is broadcast to be heard by all participants, including those not wearing the HMDs. (Further details of the technical aspects and design of the CXR system are summarized in our forthcoming publication.)

### 4.3. Participants

We recruited roughly a third of our total participants through the RI community groups. With the help of the university community engagement liaison, we reached out to the Residents Association, the Seniors Association, the Disabled Association, the Visual Art Association, the parent's group, and the two other climate-change-oriented groups on RI.

To broaden participation, we also printed and posted paper copies of our recruitment poster in the local library, the Senior Center, and the local art gallery. This process helped us to recruit another third of the participants. Finally, we contacted the local island news blog. In total, 106 people signed up to participate in our 6 scheduled rides.

The majority of the participants (80.8%) were above 45 years old (Figure 5E), and 72.9% of them had been living on RI for more than 10 years (Figure 5F).

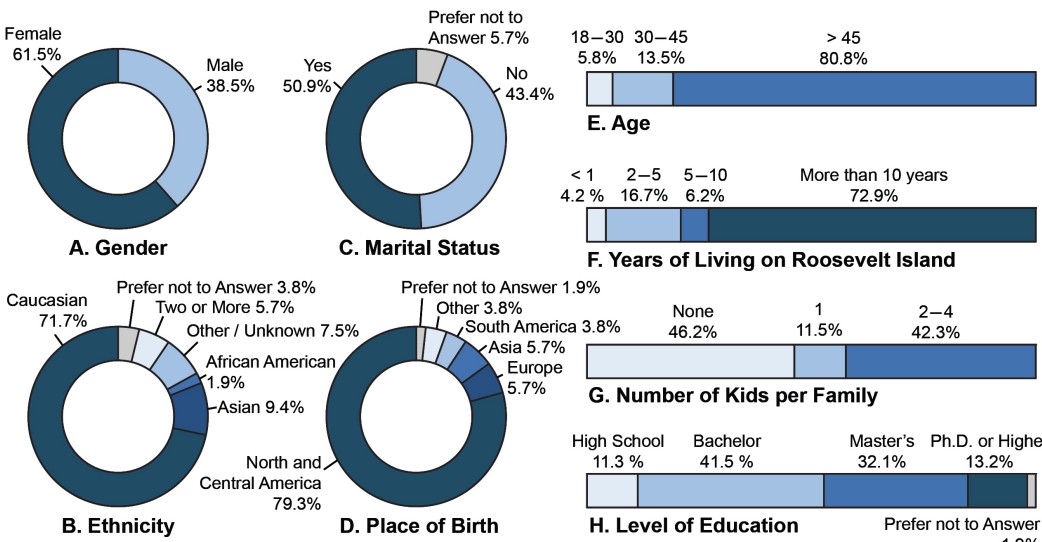

**Figure 5.** Demographics of participants.

### 4.4. Experience

During the second week of August 2022, we conducted six focus groups with 8–15 participants each, with $N = 74$ community members total. All the interactions were audio and video recorded through two action cameras.

## 5. Data Analysis

We conducted thematic coding of the audio recording from the focus groups, behavioral observations of the video recordings of the participant experiences during the bus rides, and quantitative analysis of survey data.

### 5.1. Thematic Coding of the Focus Groups Transcripts

All recordings from the focus group were transcribed by automated software—Otter.Ai (http://otter.ai (accessed on 2 March 2023)) and transferred to a collaborative, cloud-based qualitative analysis tool, Dedoose (http://dedoose.com (accessed on 2 March 2023)). The transcripts were analyzed by four members of our research team using thematic analysis techniques [62] . The team iteratively moved from line-by-line open coding and rounds of collective sense-making to create a common terminology for the application of the codes among all team members. This process facilitated our understanding of the themes emerging from the data. Our final inductively determined codes represented the full array of themes from the focus groups:

1.　Facilitation of the focus group through ice breakers and background information, such as the participants' ties to RI and their prior knowledge of climate change and its effect on RI;
2.　Feelings evoked by the ride;
3.　Participant's awareness of climate change;
4.　Participant's responses to the experience;
5.　Strategies in different levels.

Each of the above parent codes possesses a set of child codes as elaborated in the code application table in Figure 6, focusing on its dimensions. For example: <strategies in different levels> includes codes themed as <mentions of authority>, <power>, <similar situations experienced>, <strategies discussed>, <community level>, <other strategies>, <policy and systemic level>, and <things that can go into the community resiliency plan>.

| CODE | TAG | Group 1 pre G1A | post G1B | Group 2 pre G2A | post G2B | Group 3 pre G3A | post G3B | Group 4 pre G4A | post G4B | Group 5 pre G5A | post G5B | Group 6 pre G6A | post G6B | Group 7 pre G7A | post G7B | Totals | |
|---|---|---|---|---|---|---|---|---|---|---|---|---|---|---|---|---|---|
| Ice Breakers | Years of living on the island | 11 | 0 | 0 | 0 | 10 | 0 | 7 | 0 | 15 | 0 | 7 | 0 | 0 | 0 | 50 | 107 |
| | Things I like to keep from flooding | 8 | 0 | 7 | 0 | 13 | 0 | 9 | 0 | 10 | 0 | 10 | 0 | 0 | 0 | 57 | |
| Evoking Feelings | Feelings | 0 | 0 | 0 | 0 | 0 | 0 | 0 | 0 | 5 | 0 | 0 | 0 | 0 | 0 | 5 | 36 |
| | Feelings of hopelessness | 2 | 0 | 1 | 1 | 0 | 4 | 3 | 9 | 2 | 0 | 4 | 2 | 1 | 2 | 31 | |
| Awareness of Climate Change | Actionable ideas | 0 | 2 | 0 | 0 | 6 | 8 | 1 | 5 | 0 | 0 | 9 | 0 | 2 | 1 | 34 | 161 |
| | Existing knowledge of climate change | 6 | 0 | 7 | 3 | 21 | 0 | 4 | 0 | 12 | 0 | 0 | 0 | 1 | 0 | 54 | |
| | bring awareness of climate change | 0 | 5 | 0 | 8 | 0 | 6 | 0 | 8 | 0 | 7 | 0 | 4 | 0 | 1 | 39 | |
| | How climate change can affect RI | 3 | 5 | 7 | 2 | 0 | 0 | 2 | 1 | 4 | 0 | 8 | 0 | 1 | 1 | 34 | |
| Responses to the Experience | Was the system useful? | 0 | 1 | 0 | 3 | 0 | 0 | 0 | 0 | 0 | 5 | 0 | 0 | 0 | 8 | 17 | 72 |
| | The situational qualities | 0 | 3 | 0 | 7 | 0 | 5 | 0 | 4 | 0 | 4 | 0 | 0 | 0 | 2 | 25 | |
| | The immersive qualities | 0 | 1 | 0 | 4 | 0 | 0 | 0 | 0 | 0 | 1 | 0 | 1 | 2 | 11 | 20 | |
| | The communal qualities | 0 | 0 | 0 | 1 | 0 | 1 | 0 | 1 | 0 | 5 | 0 | 1 | 0 | 1 | 10 | |
| Strategies in Different Levels | Mentions of authority | 4 | 5 | 1 | 2 | 4 | 5 | 7 | 7 | 4 | 1 | 3 | 0 | 1 | 0 | 44 | 255 |
| | Past experiences | 2 | 0 | 6 | 1 | 6 | 1 | 3 | 0 | 2 | 1 | 7 | 1 | 8 | 0 | 38 | |
| | Strategies discussed | 1 | 2 | 0 | 6 | 8 | 0 | 3 | 0 | 0 | 0 | 1 | 1 | 3 | 3 | 28 | |
| | Community level | 0 | 1 | 0 | 3 | 0 | 1 | 9 | 5 | 12 | 1 | 6 | 5 | 0 | 0 | 43 | |
| | Other strategies | 0 | 0 | 0 | 1 | 0 | 3 | 3 | 3 | 1 | 0 | 1 | 0 | 1 | 0 | 13 | |
| | Policy and systemic level | 0 | 1 | 3 | 2 | 3 | 1 | 6 | 5 | 7 | 1 | 3 | 3 | 1 | 1 | 37 | |
| | Add to the Community Resiliancy Plan | 3 | 1 | 0 | 3 | 1 | 3 | 6 | 2 | 15 | 5 | 9 | 3 | 0 | 1 | 52 | |
| **Totals** | | 40 | 27 | 32 | 47 | 72 | 38 | 63 | 50 | 89 | 31 | 68 | 21 | 21 | 32 | 631 | 631 |

**Figure 6.** The code application table from our Dedoose thematic analysis processes shows the number of mentions of each code in the focus group's transcripts.

After the codes were standardized, coding was equally distributed among all researchers on the team such that there was an overlap of 20 percent of the transcripts, common to all, for standardization and sensemaking.

### 5.1.1. Behavioral Observations of the Bus Ride Videos

Our research team analyzed videos of the ride in collective and individual video-watching sessions. Each team member watched one day, wrote their initial thoughts, and shared them in a group meeting. We discussed recurring behaviors and how they correspond to the themes from the focus group's transcripts. We created a list of things to observe and then switched days. That way, each video was watched and analyzed by at least two different team members, and the results were discussed in three team meetings. Finally, we noted three main behavioral reoccurrences related to the immersive embodied qualities, the situational aspects, and the communal ones.

### 5.1.2. Quantitative Analysis of Survey Data

The surveys covered the participant's demographics, their subjective feelings on the topic of global warming and flooding, and their assessment of the effectiveness of the immersive bus ride. The collected data were quantitatively analyzed to gauge the impact of the CXR system.

From our 74 participants, we collected 60 pre-ride survey results and 53 post-ride survey results. (Not all of the participants filled in the forms. Additionally, 7 participants filled in the pre-ride survey but did not fill in the post-ride result.) The quantitative analysis is based on the responses from the participants who submitted both pre-and post-ride surveys. We used both the quantitative results and the qualitative responses from the transcripts as the basis for our findings.

## 6. Results

We coded 161 excerpts that included mentions of climate change; 54 were about existing knowledge or past experiences, and 34 were about climate change effects on Roosevelt Island. Among those, 39 excerpts were specific responses to how the CXR system changed their awareness of climate change. For example, one participant said: "*You brought an awareness that even though we've watched all of this on TV and heard about it through conversations, this really brings you face to face with the tragedy that's coming.*" (G6B).

### 6.1. Impacts of CXR on the Community

The responses varied from positive feedback to suggesting other venues to share the experience to mentioning a specific loved scene. Most people who thought the ride was effective also felt responsible for making it more noticeable to other people on the island. "*I hope this valuable exercise will transform into something that Roosevelt Island can see*", said an Emergency Manager who took the ride (G6A).

### 6.1.1. Education and Awareness

The post-ride focus groups and participants' responses showed that the experience was successful at bringing education and awareness, especially compared to other government-led activities. For example, "*The information in the city communication pamphlets is not presented sufficiently*", and after taking this ride, "*I look for my resources, and they're right here*" (G6B). The same idea was more explicitly expressed by another participant who said: "*...they always give us these numbers, but nobody really contextualizes them.*" (G7B).

The surveys included questions that helped us identify if the ride changed participants' perceptions of climate change. We asked for their personal opinion on whether the world's climate is changing and whether climate change would affect Roosevelt Island specifically. The graphs in Figure 7A,B show that all participants agree that the climate is changing and Roosevelt Island will be affected. However, for the questions about their ability as individuals and as a community to fight climate change, there is a decrease in the number of "Don't Know" answers in the post-ride results, which suggests that the CXR ride affected those who were unsure. As shown in Figure 7C,E, more participants are certain that they can (64.2% to 73.6%) and they know how to (58.5% to 66.0%) help fight climate change after the ride. Few remain uncertain about this, and the number of "Don't Know" responses is

decreased from 34.0% to 24.5% on whether they can contribute and decreased from 28.3% to 22.6% on whether they know how to contribute.

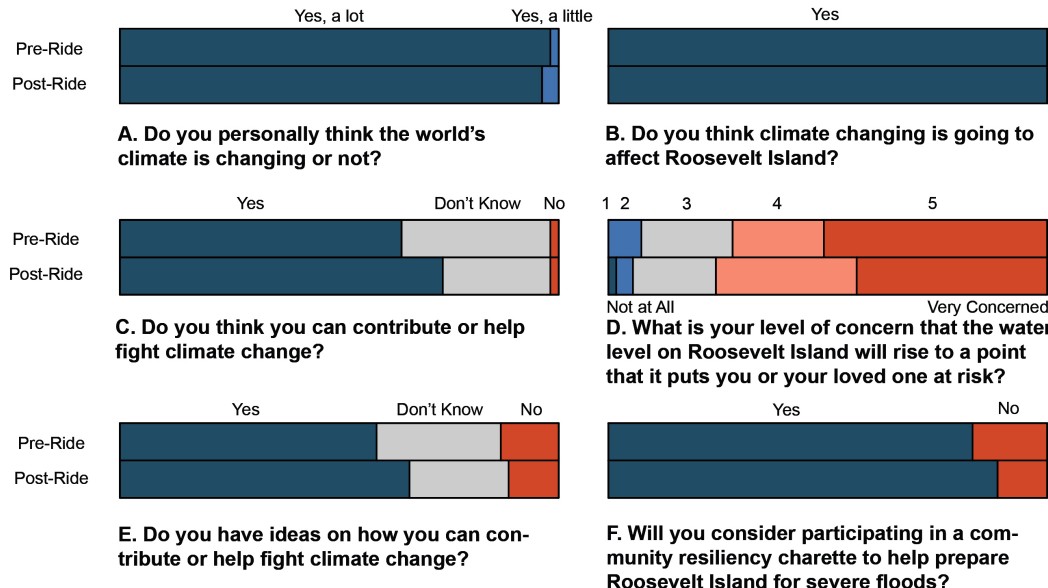

**Figure 7.** The change of answers in the post-ride survey.

This question was followed up with a request for examples of some actions that can be taken to fight climate change. In the post-ride survey, we collected 44 examples relative to 38 examples in the pre-ride, indicating an increase in the ability to imagine and think proactively. Moreover, the ideas in the post-ride focus groups were more elaborate and specific. Similar changes were found regarding the actions that can be taken as a community; the number of ideas increased from 57 to 63, with more elaborate suggestions in the post-ride survey. More of them considered participating in a community resiliency charette to help prepare Roosevelt Island for severe flooding (Figure 7F). The number of positive answers increased from 83.0% to 88.7%. Finally, when directly asked, "*In your opinion, was the IMMERSIVE XR TOUR effective?*" 35.8% answered "Somewhat Effective" and 60.4% answered "Very Effective" (Figure 8A). This shows that our ride and information effectively raised awareness and supplied some actionable ideas. A number of participants were more concerned about the water level rising and felt less prepared for a flood after the ride. Figure 7D shows that although fewer participants rated 5 out of 5 for their level of concern (50.9% to 43.4%), there is a higher number of participants who raised their rating to 4 out of 5 (20.8% to 32.1%).

### 6.1.2. Bringing Community Members Together

The experience brought community members together and provided a safe environment to express fear and hopelessness. Overall, we coded 36 mentions of hopelessness facing climate change, 24 were in the post-ride focus group, and only 12 were in the pre-ride group. Some statements clearly stated, "*I'm feeling very helpless, helpless, and hopeless. Because, you know, we can do things like buying electric cars, compost our garbage, you know, the little things... But in the grand scheme of things, it just feels like it's going to come. It's coming, and there's nothing we can do about it.*" (G6B).

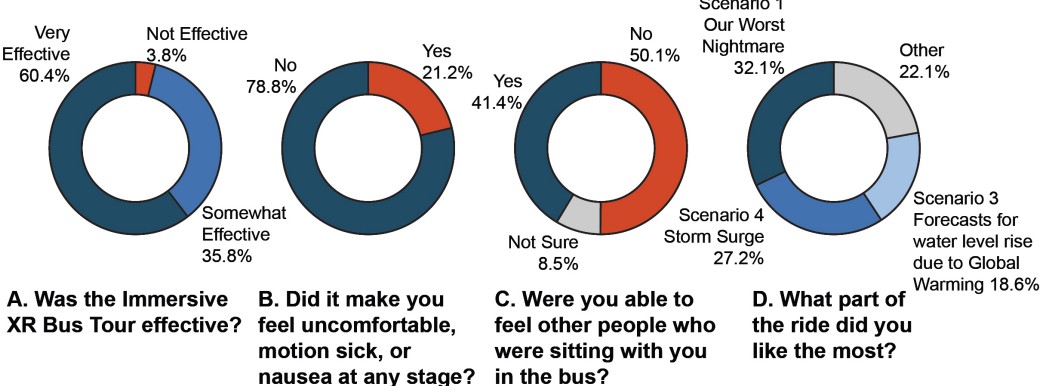

**Figure 8.** Participants' view of the CXR system as reported in the survey.

While most of the discussion in the pre-ride was general, there were more descriptive and specific fears in the post-ride group. The experience enabled them to give names to things they experienced in the past. For instance, the script included what storm surges are going to look like, providing the scientific terminology. In the post-ride focus group, participants used the same terminology to describe their concerns: "*It seems the storm surge could come at any time because that's really, you know, with this gradual rise in sea level, the storm surge, like in Superstorm Sandy, they can come anytime*." (G3B). Overall, participants were happy to be informed and talk about local issues that concern them in a safe environment: "*I appreciate that you're opening the dialogue so that it's a community awareness...People need to talk and be aware that you're raising the consciousness of this, I think, is a great thing*." (G3B).

### 6.1.3. Identifying Local Priorities

In our community engagement process, we learned that the community officials were worried about scaring people away from living on the island. Initially, we were alarmed that we might have provoked the exact reaction they feared. "*I was really feeling that there is nothing more that we can do*." said one participant, exclaiming that "*maybe we should give up and try to find where we'll be able to live, away from the shores*." (G3B). However, analyzing the transcripts further, we found that hopelessness and fear evoked creativity, ideas that people would not have thought about without the stimulating experience of danger. One of our participants expressed the idea of lifeboats: "*Obviously, you can't helicopter everybody from the island. So what about boats? They are probably the only reliable way. And you will need a lot of them. I suggest surrounding the island with inflating lifeboats like on those cruise ships*" (G3B).

We discovered a repetitive pattern in the transcripts: devastating insights or feelings of hopelessness brought up by the experience were usually followed by creative ideas and locally oriented solutions. The following are examples:

- Concern: "*There are a lot of senior residents on the island*." Actionable idea: "*Get people to be buddies with some of the older residents who live alone*." (G3B).
- Concern: "*The one thing I took away from it is how useless cars are going to be*." Actionable idea: "*Have a plan with your family. How are you going to connect? The meeting place where you're going to go? Keep your documents together, list of your medications, emergency numbers, all that kind of stuff*." (G5B).
- Concern: "*Not everybody knows how to understand alerts on their phone*." Actionable idea: "*When I grow up we had an alarm system... It's a small island, I imagine you could do something like that here...*" (G5B).

### 6.1.4. Enhance Ownership and Accountability

Participants debated who is responsible for taking action to fight climate change. On the one hand, we coded "they" mentioned in the focus groups transcripts, with pushback to take individual responsibility and blame different levels of governance or authorities: (federal, national, or municipal), industry (oil companies, fashion industry), the city of New

York, the island operation corporation, or even the building's management. On the other hand, we found mentions of "we": people who were actively looking to find resources within the community or act themselves.

"*...They don't really take seriously or even notice us on the state level.*" (G1B). Or, "*It's more of a city or a world issue. Actually, carbon, or, like reducing the emission gases is something that is global...*" Some claimed the transportation authority should take responsibility, and others shared their frustration with the local operation corporation.

"We" mentions included proactive ideas of leveraging the community's power and resources to lead the local management to action. For example, encouraging us (the researchers) to create a pressure group of the participants "*I would strongly encourage you to keep us in the loop... if we can pressure them to do some of the things that you all think might need to be done...*" (G3B).

Ownership and accountability were expressed through the participant's concern about who would be responsible for different tasks. They thought backup from the governance is crucial, but some strategies at a community level could advance independently. One practical idea was to regard each building as a sub-community. "*I think on the island it is actually quite straightforward to communicate things because you only have like these big apartment complexes ... 12 or 13 of them, and they actually communicate quite frequently with their tenants. So that would actually be quite easy to reach*" (G1A). Another idea was to build a committee of experts with different skills (such as front-line workers) to guide residents; they mention these willing people previously proved to be more helpful than certified experts. They also suggested creating a list of vulnerable populations, such as seniors or the disabled, and making sure they are prepared and taken care of.

We noticed a change in attitudes after the ride when climate change became more realistic. There were more expressions of agitation about inaction at all levels. The main frustration was that no concrete actions were taken: "*There are attempts in community engagement, but that's not enough. The buildings themselves should have active plans, and they should be talking with the residents.*" (G3A).

Despite a high degree of community accountability, participants thought governments should take ownership and accountability to fight climate change. "*Planting flowers is more practical, but then you need to understand where the world is going ... it's not going to be helping planting trees ...*" (G1B). They thought such efforts as movies, recycling, or gardening were not enough. They wanted to know concrete actions and learn how these could help Roosevelt Island. They expected the operating corporation to publicize the intensity and level of near-future danger that climate change might hold for Roosevelt Island.

The unknown evacuation plan remained the primary concern both in the pre- and post-ride responses. Participants described how, to their knowledge, there was only one evacuation site for the 12,000 people on RI, and it was poorly connected. They mentioned receiving random information regarding evacuation during Sandy and claimed that they lack information, knowledge, and expertise to act during an emergency. There seemed to be a lot of ambiguity around responsibility and timing. "*And nobody on Roosevelt Island seem to know about any of that stuff... whoever's in charge needs to do a much better job of letting people know what's going on.*" (G3B).

Some actionable systematic-level strategies were suggested, for instance, to offer climate-change training repeatedly "*you need to make sure that annually, you review it and do it because the phone numbers change and people change. And that's something that needs to be refreshed on a regular basis*" (G3B). Additionally, they asked to be informed about the emergency preparedness plan, suggested having flood drills and flood alerts, and further creative ideas (elaborated in the draft community resiliency plan in the Appendix A).

*6.2. The Advantages of CXR for Climate Change Community Engagement*

On the whole, people responded positively to the CXR experience and seemed engaged in the topic of climate change that the CXR experience focused on. In our thematic analysis, 72 excerpts referred directly to the qualities of our CXR; 17 commented about

the usefulness in conveying the risks of climate change; 20 talked about the immersive embodied aspects; 10 highlighted the communal aspects; and 25 related to the situational qualities (see Figure 6). Most responses were positive, directly expressing the experience's effectiveness. Five participants openly critiqued aspects of the ride, with such comments as the following: "*the specific levels of water that you mentioned is sort of like well, yeah, that's speculative. So I could speculate too.*" (G2B). "*I think that you've been carried away by the artistic possibilities of the medium*" (G3B).

There were some critics related to technical problems. Naturally, we had more technical issues in the first rides, which improved as our team became more proficient in operating the system. Participants complained about issues such as starting the app on time, orientation within the virtual world, or low battery warnings.

The post-ride surveys also reflect technical complaints, elaborated in Figure 8B. Although 78.8% of participants did not feel motion sickness or nausea at any stage during the ride, 21.2% indicated some uncomfortable experiences.

6.2.1. The Immersive, Embodied Aspects—Making Climate Change Feel Real

The immersiveness of the experience was created through a 360-panoramic video of the island that people saw through VR headsets on the moving bus. Off-the-cuff responses, such as "*Whoa, whoa, this can really happen,*" (G3B) or "*It's scary. It's a little shocking, especially for young moms, and recent homeowners*" (G3B) showed us that the experience was effective in making climate change feel real. "*The value of the ride is . . . the way you put it in-your-face; that's when it becomes more real.*" (G7B). One of the participants clearly stated the difference made by the immersion by saying, "*...many times they just read those things, and I'm like, why aren't people running in the streets asking to stop those crazy things that are happening? And I guess it's because we don't have enough of these (referring to the CXR). So I'm saying bring it on! Again! With the anxiety, with the warnings with everything*" (G3B). One participant said, "*After going down from the bus, I feel like my feet are wet.*"

One of the common challenges in creating VR rides is motion sickness and nausea. To our surprise, there were a few complaints of nausea and car sickness. One person out of seventy-four said, "*I did feel nauseous. You know, especially with the visuals sort of not following the bus movement.*" (G2B). In the survey, there were more reports about nausea and motion sickness. However, the majority of participants seemed not to experience discomfort. We believe these encouraging results are related to the system's qualities of good GPS accuracy, the low latency between the bus movement to the in-virtual-world action, and the affordance for participants to change their orientation by using a VR joystick.

The video analysis shows how even though each participant was in their own virtual world, the immersive, embodied qualities of the ride still worked to create a communal experience. At many points along the route, we identified shared bodily responses to views in the headsets. For example, Figure 9 shows how people moved their heads in the same direction when they noticed the whale or other spectacles in the virtual world. At those moments, there were usually many vocal reactions and shots, which evoked conversation among the participants.

6.2.2. The Situational Qualities—Bringing Climate Change Closer to Home

Being situated meant that the participants were physically located in the same place in the real world and the digital twin, with a narrative built according to specific places and statistics about Roosevelt Island. People repeatedly talked about how seeing familiar places, such as the school, the library, or their favorite playground underwater, brought the intangible political climate crisis "*a little closer to home.*" (G6B) (Figure 10). From an abstract, remote problem, it becomes related to something familiar: "*What you're doing is very important because it brought back Sandy... I mean, a flood on Main Street like that...*" (G6B). The ability of VR to change the neighborhood they know contextualizes climate change and makes it more tangible. "*It was really helpful to seeing the actual place. It kind of puts things in perspective, not just like a random report.*" (G7B).

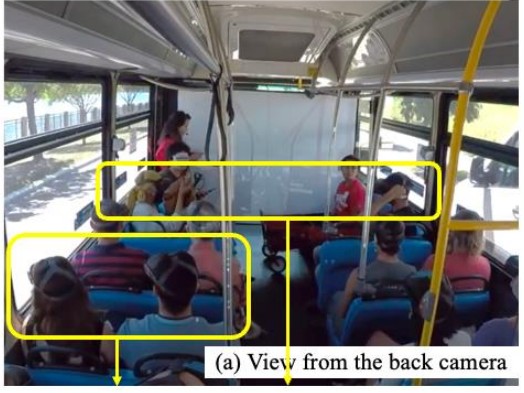 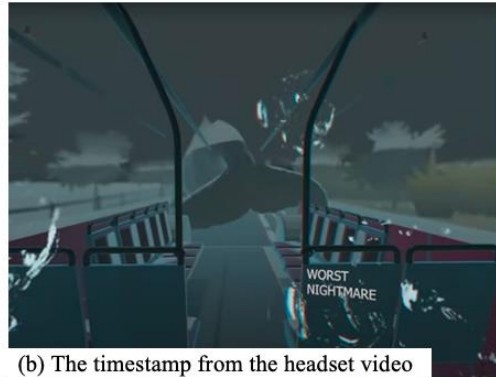

(a) View from the back camera        (b) The timestamp from the headset video

Participants staring simultaneously at the whale's direction      Participants showing bodily response when noticing the whale        The Worst Nightmare  scene

**Figure 9.** Participants embodied responses.

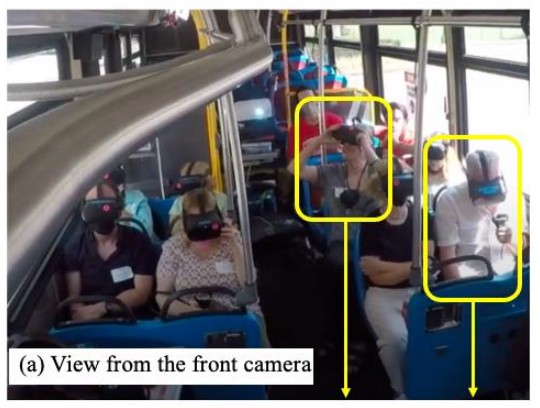 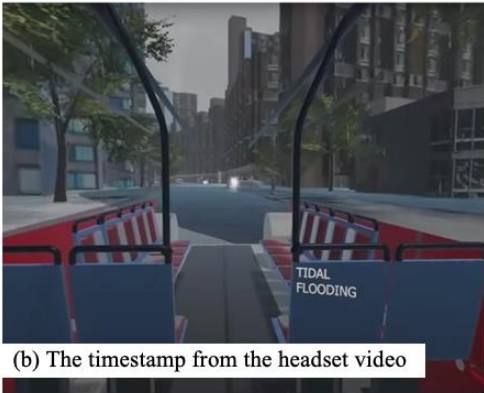

(a) View from the front camera        (b) The timestamp from the headset video

A participant pointing to her apartment      A participant estimating the water level near the local library        The Tidal Flooding scene

**Figure 10.** Video analysis: participants identifying and physically responding to a familiar location.

However, some thought that the bus ride's geo-location was redundant. "*I felt, frankly, that riding on the bus was almost superfluous to the experience...I mean, it didn't matter to me; the VR was all-encompassing. The fact that it was sitting on the bus or not didn't make much difference.*" (G2B). On the other hand, some appreciated the situated experience and the way it enabled them to be in the same place at different periods in time—going back in history to Hurricane Sandy and then fast-forwarding one hundred years into the future: "*When we stopped, and we were looking through the virtual reality, then lift the headset so that we're looking at the area we're sitting in. And then we say, oh, that's the difference.*" (G6B).

In the ride videos, we found moments of communality, where people identified familiar places and physically reacted to them by either pointing or speaking out loud. For example, in Figure 9, one participant identified their apartment, and another pointed in concern to the local library.

### 6.2.3. The Communal Aspects—Turning the Spotlight to the Community Resources

The experience brought together people from the same community. Some had been neighbors for years, and some knew each other from sporadic meetings or events. They all were familiar with the location and the bus they were sitting in. Due to the communal aspects of the experience, many of the conversations started from this point of familiarity. This was helpful in diving deeper and faster into the subject. Additionally, many of the actionable ideas came from past experiences of the community. Some of the excerpts from

the focus groups related to those aspects. For example, someone said: "*I'm just really enjoying being with everyone and bringing the subject to the foreground*" (G3B).

During the bus ride, although each participant had their own view in their own headset, they all listened to the same soundtrack, could hear each other, and felt the presence of others in the same space. There were also scripted moments where we instructed everyone to take off their headsets and look around. Some participants appreciated the communal aspects more than the immersion and chose to keep their headsets off. "*I thought it was more successful personally when I had the headset off, and I was just sitting there listening to you speak. I think it would work equally as well. If you just had a projector on the front screen and all the passengers watched that screen. And that was because also what's important is your information, the things you're saying. And when you have that headset on you're pretty distracted.*" (G7B). Interestingly, although the VR HMDs block the participants from seeing others on the bus, the pie chart in Figure 8 shows that 41.4% of participants were able to feel the presence of other people near them.

From the video analysis, Figure 11 shows the moment that people removed their headsets at the end of the ride and acknowledged the other people sitting next to them. At this moment, there were usually sighs of relief, laughter, and lively conversation. It was a moment of community building—the shared experience intuitively connected them, creating a feeling of a shared future. Moreover, the ride ended near the local hospital and an iconic historical residential building, which were flooded during Hurricane Sandy. As shown in Figure 11, this location was rendered in the movie as the "After a Flood" scene, with debris and fallen trees all around. The specific location, and the way the movie portrayed past experiences, turned participants' attention to the chronically ill patients of the hospital and the historical monument. Both were harmed by previous floods and are at the most immediate risk of being flooded again. This combination turned their attention to the precious things in their community and the resources that are available, which were mentioned by the participants when taking off their headsets at this point.

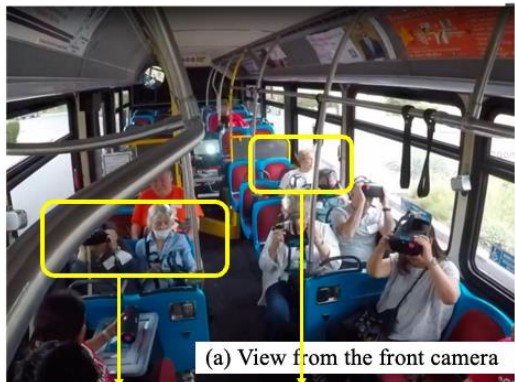 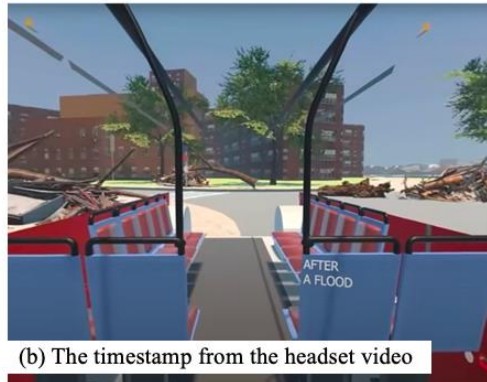

(a) View from the front camera  (b) The timestamp from the headset video

Participants showing signs of relieve at the end of the ride

Participant checking their surroundings to validate reality

The After a Flood scene

**Figure 11.** Being in the same place and going through the shared experience increased the communal shared future aspects.

## 7. Discussion

Following previous calls for innovative methods for engaging people on complex social and environmental issues [13,63,64], our primary goal was to understand how CXR experience can probe and promote community awareness and resiliency of climate change. We deployed and evaluated our CXR bus ride, which offered a group of people from a coastal community an immersive environment composed of shared 360-panoramic video in VR, situated and geo-located on the road. We hypothesized that such an experience would be more effective than prior methods due to its combination of digital twin-based and real-time visualization, situational experience, and communal qualities. We conducted

seven bus rides with 8-15 participants in each, including pre- and post-ride focus groups. We recorded and transcribed the deployment process and thematically coded our data.

Like previous work on climate change community engagement, the main achievements of our CXR are as follows: bringing stakeholders together to develop locally-appropriate solutions [18,19]; promoting education and awareness about climate change [1]; identifying local priorities for local and specific climate-change-related issues [20,21]; and enhancing ownership and accountability of climate-change-mitigation initiatives Krasny and DuBois [22]. Moreover, by providing a safe environment to share feelings of hopelessness and fear, the CXR engagement helps evoke creative thinking and lead to innovative and action-oriented ideas. Additionally, after the ride, participants showed willingness to take action and participate in further climate change activities.

On top of that, our CXR contributes to the existing knowledge in three dimensions:

1. It suggests an empirical contribution, with results from the first-of-its-kind, real-world deployment of the CXR system with participants from the community.
2. It offers a methodological contribution of a community engagement protocol, the unique opportunity to use mass transport vehicles as a venue to engage with communities in their familiar and most precious environment.
3. It makes a practical contribution in the form of a draft of a community resiliency plan. The draft in the Appendix A was collected through the focus group engagement protocol and can serve as a base for future development by the community.

### 7.1. Immersiveness Makes Climate Change Feel Real

Previous research showed that making climate change more tangible and proximal improves community engagement [8,10,11]. Our CXR, immersive nature, combined with the situated aspect of the bus ride, took this claim a step forward. Like Breves and Schramm [14], who showed that increased immersion highlights proximal issues and enhances pro-environmental behavior, our participant's reactions and feedback showed that the CXR experience effectively conveyed the potential dangers of climate change, successfully created a personal connection to the issue, and promoted a sense of community responsibility. However, while they conducted an experiment in the lab, our CXR took the participants to the on-site location.

### 7.2. Situational Qualities and Geo-Location Brings Climate Change Closer to Home

Our communal CXR system is a first-of-its-kind development in immersive extended reality (XR), offering an unprecedented combination of geo-synchronized and communal XR experiences. Prior traditional designs of on-road VR systems, such as Holoride and VR-Coaster, have predominantly focused on rendering the visual perspective of an individual passenger within the moving vehicles. In contrast, shared VR experiences systems, such as shared room-scale virtual and mixed reality storytelling for multi-people, developed by Herscher et al. [34], position the audience in a physically co-located studio setting. Our significant innovation is to combine these two distinct approaches, creating a shared yet reality-correspondent experience through our cutting-edge CXR system.

Experiential learning and place attachment have been proven to activate community-level awareness and action [31]. Leveraging this knowledge, our narrative included familiar places and past events related to the community. We found similar responses to Calil et al. [42], who found that familiarity with the locations in VR facilitated discussion on immediate issues for the community and triggered stronger emotional reactions. People who identified their home, the local school, or the library responded physically (pointing), vocally (shouting), and emotionally (mentioning it again in the focus group). The situated aspect of our CXR narrative helped bring the issue of climate change closer to home, making it more relatable and tangible, allowing them to see familiar places in their community in the past or future.

### 7.3. Communal Experience Highlights Existing Community Resources

Hsu et al. [36] and Markowitz and Bailenson [37] showed that participation that allows communities to feel connected emotionally to matters of concern by sharing their first-hand or relatable experiences is more effective. In our case, the communal aspects helped participants ideate specific and tailored solutions for their community. Like Stone et al. [38], we provided opportunities to share ideas about action, mitigation possibilities, and adaptation. Additionally, the focus groups and ice-breaker activities created a sense of familiarity among the participants and helped them to dive deeper and faster into the subject. Using the familiar bus ride as a community-building activity highlighted existing community resources. During the bus ride, the shared experience of listening to the same soundtrack and feeling the presence of others in space helped to create a sense of community and connection. The union was created not only because of the forecast future but also through the technical and physical difficulties they experienced together, creating a sense of shared destiny and friendship.

### 7.4. Toward a Community Resiliency Plan

As with previous research, we showed how technology emphasizes collaboration among government and community stakeholders [14,42,43,53,65,66]. Our unique contribution, in this case, is the unique venue of the bus. Our engagement process suggests a unique opportunity to use mass commuter vehicles to engage with communities during their daily commute—the most important locations. A more advanced version of a CXR experience could channel the time for commuting into a valuable contribution to society.

Our participants had different perceptions of who should take responsibility for mitigating climate change. Some proactively offered ideas to leverage the community's power and resources. Some encouraged us to pressure groups of the participants to influence the local management to take action. Overall, the participants recognized the need for collaboration and awareness-raising efforts, including the need for the government and community to work together.

As a practical contribution to this research, we collected the participant's suggestions into a draft community resiliency plan, added to this paper in the Appendix A. The draft introduces the concept of growing circles of responsibility for climate resiliency. It is composed of local knowledge based on past experiences, communal thinking, and creativity evoked by the immersive bus ride. It is collected and edited by the researchers but comes from the people who know the place best. It is not based on any other community resiliency plans or wider knowledge. To respond to participants' concern for collaboration and trust building with the local government, we plan to share it with the participants and local officials who took part in the experience to obtain their feedback and create more collaboration opportunities to lead to a formal community resiliency plan.

### 7.5. Limitations

As noticed by [42,43], renderings of long-term adaptation solutions that balance realistic options and visionary concepts are challenging, as stakeholders often do not agree on the details of the final visuals or the narration accompanying them. We faced similar limitations in our processes as we described in Section 4. As mentioned in the results, some of the participants expressed their discontent with the inaccuracy of our visuals and story.

We will elaborate on the technological challenges in another forthcoming work focused on the CXR system. Some issues include GPS latency, failure of the VR headsets, or orientation challenges of the 360 video. Beyond these technical limitations, we believe it would be useful to explore the possibility of creating in-car, position-based, real-time rendered avatars for a multi-user VR experience. Allowing participants to see each other on the bus can improve their sense of presence and commonality.

However, our main limitation is related to the participant's diversity. Due to the dependency on the mass transportation system, we were able to conduct rides only on weekdays and during the morning and early afternoons. The result was that our group

of participants included mainly senior citizens. We were able to balance the age group to some extent by recruiting one student group and by conducting a Friday ride. However, future research should focus on increasing diversity.

Another general limitation of our data collected is that our system is multi-aspect (situational, visual, and communal) and multi-component (bus, visualization, headsets, university facility used for the focus groups, researchers); as such, it is difficult to separate out which response and finding can be attributed to any particular characteristic or feature of our system.

### 7.6. Future Applications to Promote Community Awareness and Engagement

Improvements to the system could allow more people on the bus and allow for interactive engagement with each other. We also suggest giving more scripted opportunities to remove the headset and engage in collaborative activity or discussion. Deployment of alternate technology for greater access and awareness would increase the inclusivity of the experience, for example, creating QR codes on the bus for people to scan and see the 360 videos on their phone or QR codes in specific places on the island. Additionally, we suggest doing more rides with other age groups, specifically more parents. Diversifying participation of stakeholders by scheduling specific rides with officials and decision makers. Teasing out causal relationships using experimental conditions by having more groups in the lab simulators to be able to compare with the situated geo-location.

### 8. Conclusions

Our realistic visualizations from the DT of Roosevelt Island, used on a local shuttle bus, provided an in situ experience of being in a place while using VR and allowed multiple users from the community to share the ride and discuss it in pre- and post-focus groups. The CXR is a first-of-its-kind empirical contribution to inspire action. It offers a proof-of-concept of a large-scale community engagement process featuring simulated communal experiences leading to creative ideas for building a bottom–up community resiliency plan.

Like previous research on community engagement and climate change, we showed that technology is a powerful tool to bring people together, educate and raise awareness, help identify local priorities, and enhance ownership and accountability. Our CXR brought people together to discuss climate change in a safe and enabling environment to express fear and hopelessness.

Adding to the research in the field, our CXR has original advantages. While previous communal VR experiences were conducted in the lab, our CXR took the participants to the on-site location. While previous on-road VR was used by individuals, our CXR enables multiple users a common and synced experience. Our significant innovation is the combination of these two distinct approaches, creating a shared yet reality-correspondent experience. Our results show that the unique qualities of our CXR of immersive, embodied, situated, and geo-located contributed to the effectiveness of the experience in three ways: The immersive, embodied aspects made climate change feel real. The situational qualities brought climate change closer to home. The communal aspects turned the spotlight to the community resources.

Moreover, by simulating a physical emergency state, the CXR experience helped translate feelings of hopelessness into creative, actionable ideas. The CXR created a sense of urgency and empathy by connecting the participants to their local community and the physical effects of climate change. Another important advantage of our CXR is that it led to creative ideas without putting anyone in real danger. Although people were faced with disaster only in the virtual world, it led to creative ideas in the real world. This finding exemplifies how CXR can be a powerful tool to lead to innovations and creativity.

The study provides evidence that CXR experiences can be a powerful tool for building community resiliency and encouraging action on climate change. It offers an empirical contribution of a community engagement process on a large scale, a methodological contribution of an engagement process using public transportation, and finally, a drafted

community resiliency plan as a practical contribution (see Appendix A). This work points to the ways that the urban data and digital models of the automated city might be applied to influence community awareness and motivation to address the world's most pressing problems.

**Author Contributions:** Conceptualization, S.Y.-A. and W.J.; methodology, S.Y.-A., S.J. and W.J.; software, Y.Z. and R.H.; validation, S.Y.-A., S.J., Y.Z., R.H. and W.J.; formal analysis, S.Y.-A., S.J., Y.Z. and R.H.; data curation, S.Y.-A. and S.J.; writing—original draft preparation, S.Y.-A. and S.J.; writing—review and editing, S.Y.-A., N.M. and W.J.; visualization, S.Y.-A., Y.Z. and R.H.; supervision, S.Y.-A., N.M. and W.J.; project administration, W.J.; funding acquisition, W.J. and S.Y.-A. All authors have read and agreed to the published version of the manuscript.

**Funding:** This work was sponsored by research funding from Tata Consultancy Services.

**Institutional Review Board Statement:** Not applicable.

**Informed Consent Statement:** Informed consent was obtained from all subjects involved in the study.

**Data Availability Statement:** Data is contained in the study.

**Acknowledgments:** This work was sponsored by research funding from Tata Consultancy Services and was supported by Roosevelt Island Operating Corporation [RIOC]. We especially thank Cyril Opperman, RIOC's Director of Transportation, who enabled us unlimited access to the bus, and Jane Swanson, our Assistant Director of Government and Community Relations, for her invaluable assistance in community outreach and participant recruitment. In addition, we thank Glenn Baevsky for narrating the video and for his technical support while prototyping the bus.

**Conflicts of Interest:** The authors declare no conflict of interest.

## Abbreviations

The following abbreviations are used in this manuscript:

| | |
|---|---|
| CXR | Communal Extended Reality |
| NYC | New York City |
| RI | Roosevelt Island |
| AR | Augmented Reality |
| VR | Virtual Reality |
| XR | Extended Reality |
| HMD | Head-Mounted Display |
| DT | Digital Twin |
| RIOC | Roosevelt Island Operating Corporation |

## Appendix A. Initial Ideas for a Community Resiliency Plan

This draft is built out of the conversations with the island's residents in seven focus groups that took place between 21 July to 12 August 2022. Each group participated in a communal extended reality bus ride, which immersed participants in a visualization depicting expected climate change, sea level rise, and floods on the island. Before and after the rides, we moderated focus groups that encouraged participants' creative thinking toward becoming more active in mitigating climate change effects. As a takeaway for the community, we collected key suggestions from the focus groups in this draft executive summary. This will be distributed among the participants and the island officials, as well as with broader NYC climate mitigation groups.

This is a collection of local knowledge based on past experiences, communal thinking, and creativity evoked by the immersive bus ride. It was assembled and edited by the researchers but comes from community members who know Roosevelt Island well. It is not based on any other community resiliency plans or wider knowledge. Further work with community resiliency experts and the community is needed in order to make this a viable, effective plan.

*Appendix A.1. Growing Circles of Responsibility for Climate Resiliency*

Community preparedness should include different levels of preparation over time. We combined the suggestion according to growing cycles of responsibilities: from the individual to the communal to the systemic.

Appendix A.1.1. Household Circle

- Go-bags: Keep your documents together, including a list of your medications, emergency numbers, IDs, passports, medical insurance, and so forth.
- Stay-bags: In case there is a need to shelter in place, prepare a kit with canned food, water, flashlights, batteries, plastic sheeting, duct tape, toilet paper, whistle, light sticks, blankets, etc. (these two lists, are initial thoughts that need to be improved by professionals).
- Family meet-up: Have a plan with your family. How are you going to connect? Where will be the meeting place where you will meet?

Appendix A.1.2. Building Circle

- Floor level: Check on your neighbor who shares the same floor by knocking on the door. Meet in the hallway and make sure everyone is okay and has enough supply. Make sure to keep communicating throughout the emergency. If someone is in need, reach out to the building management for help.
- Higher floors assist lower floors: In case of damage to the lower floors at higher risk for flood, be prepared to host your lower floor neighbors.
- Building information: The building's management should communicate all news and announcements through the building apps. There should be a non-electric, non-technical way to communicate information during an emergency when there is a power outage or for people who do not have access to computers, such as printouts and voice announcements.
- Building flood drills: Each building should have well-communicated and practiced flood drills renewed on an annual basis to accommodate newcomers and keep all island residents up to date with the information on what to do during a flood.

Appendix A.1.3. Community Circle

- Coordination between buildings: In the event of a flood, each building becomes a silo. However, there should be a way to communicate and coordinate activities and sharing of resources. Each building's management should appoint an employee responsible for flood preparedness who will become a part of an island-scale network that will share resources and updates with their building tenants during a flood.
- Social groups: The island has many active social groups; those could be a go-to address in case of a flood. The leaders of each social group should be informed on how to act and be responsible for communicating information to their group members.
- Flood emergency committee: Some of the island's residents, such as front-line workers, health care workers, or other relevant skills, are experts in their field. Those people have proved to be very helpful during past emergencies. The participant recommended creating a committee of willing certified experts who will be available to assist with resources and guides to the community during a flood.
- Flood preparedness advocates: Create an advocacy group out of the research participants to reach out to the island's operating corporation to advance some of the ideas brought up by the community to lead to long-term preparedness activities. This group could also be useful in reaching out to higher-level governmental authorities to allocate funds and resources to better prepare the island's infrastructure for future sea level rise.
- Flood vulnerable community members list: Create a contact list of people dependent on ventilators or living alone to reach out to them in an emergency.

- "Buddies" program: That will match younger people or families with senior citizens or other residents who live alone to make sure those are being taken care of during an emergency.

Appendix A.1.4. Governance Circle

- An evacuation plan: There should be a clear and communicated emergency evacuation plan. The content of this plan should be shared with the residents.
- Education in advance: Should start as soon as possible by reaching out to the community using relatable and everyday activities like gardening. There should be awareness-raising activities, such as informative movies, to prevent a chaotic situation in times of flood.
- Climate change training sessions: Should be offered in public locations and advertised in order to include different and varied people from the community.
- Annual flood drills: These will be performed and include all circles of responsibilities elaborated above.

Appendix A.1.5. Creative Ideas to Be Further Explored

This section includes some of the most creative ideas that might not be applicable in the near future but are nonetheless worth considering.

- Lifeboats: Can be placed around the island like those who are placed on a ship to enable quick evacuation.
- An evacuation elevator: Could be installed at the bottom of the Queensborough Bridge to enable quick evacuation.
- A digital twin for emergency control: Populate our island-scale digital twin with the list of vulnerable community members list and the Buddies program to be able to locate the people in need during a flood. Those people can be supplied with a call mechanism that will light an alarm in the digital twin when they use it.
- Flood alerts: A sound system that communicates emergencies in case of cellular network failure. An alarm system that can be heard all over the island with speakers can inform people on how to act in an emergency. This will make sure people with no cell phone, TV, or other media will receive the information too.

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
