# Peer review of "Building Community Resiliency through Immersive Communal Extended Reality (CXR)"

_mti, doi:10.3390/mti7050043_

Round 1
Reviewer 1 Report
The paper addresses the deployment and evaluation of a communal geo-located virtual reality bus tour presenting participants with floor and climate change scenarios. The purpose was to understand community member awareness and motivation to take action where the consequences of the presented climate change were closer to home.
This is a well-written paper. The research study was well-planned, developed, and executed over a year. The design and methodology of the study were sound. However, as stated in the limitations section, one of the main limitations of the study is that the demographics were not well represented, e.g., in terms of age range. Nevertheless, the study presents some insightful findings and demonstrates the effectiveness of communal XR in a contextual setting for raising awareness.
Author Response
Reviewer 1
(x) I would not like to sign my review report
( ) I would like to sign my review report
Quality of English Language
( ) English very difficult to understand/incomprehensible
( ) Extensive editing of English language and style required
( ) Moderate English changes required
(x) English language and style are fine/minor spell check required - we addressed all minor suggestions and run several proof edits to ensure there are no typos
( ) I am not qualified to assess the quality of English in this paper
|
Yes |
Can be improved |
Must be improved |
Not applicable |
|
|
Does the introduction provide sufficient background and include all relevant references? |
(x) |
( ) |
( ) |
( ) |
|
Are all the cited references relevant to the research? |
(x) |
( ) |
( ) |
( ) |
|
Is the research design appropriate? |
(x) |
( ) |
( ) |
( ) |
|
Are the methods adequately described? |
(x) |
( ) |
( ) |
( ) |
|
Are the results clearly presented? |
(x) |
( ) |
( ) |
( ) |
|
Are the conclusions supported by the results? |
(x) |
( ) |
( ) |
( ) |
Comments and Suggestions for Authors
The paper addresses the deployment and evaluation of a communal geo-located virtual reality bus tour presenting participants with floor and climate change scenarios. The purpose was to understand community member awareness and motivation to take action where the consequences of the presented climate change were closer to home.
This is a well-written paper. The research study was well-planned, developed, and executed over a year. The design and methodology of the study were sound. However, as stated in the limitations section, one of the main limitations of the study is that the demographics were not well represented, e.g., in terms of age range. Nevertheless, the study presents some insightful findings and demonstrates the effectiveness of communal XR in a contextual setting for raising awareness.
We thank Reviewer 1 for their supportive review.

Reviewer 2 Report
Overall, the paper is very interesting, addressing a pertinent topic.It is well written and structured. Although it has 24 pages, I believe readers will have a good experience analysing it. The number of pages also reflect the amount of work that has been done. All in all, my only concern is associated with the introduction, besides that, the manuscript is quite mature.
Next, some comments and suggestions for improve the quality of the manuscript are described.
- With all due respect, I believe that is too much detail on the results being mentioned in the introduction. In contrast, there is not enough information regarding the topic being address. Hence, I urge that the author rephrase the introduction, providing more detail on the importance of the topic at hands, and the use of VR for such context, and reduce the amount of detail regarding the details. (Some of these details may be re-used for the conclusion for example, which is rather small).
-- I would go as far as to say that a similar approach could be applied to the abstract.
I suggest removing the following from the introduction:
"Our effort culminated in three sets of data: videos and transcripts of the focus groups, videos of the participants during the ride, and results from 74 pre- and post-ride surveys. We thematically analyzed the data for recurring themes. In this paper, we show that although confronting face-to-face with the forecasted future evoked passing feelings of hopelessness, it was effective in improving community awareness of climate change: by bringing stakeholders together, providing information, and enhancing ownership and accountability. We argue that the immersive, embodied aspects of CXR make climate change feel real, its situational qualities bring climate change closer to home, and the communal aspects turn the spotlight to the community resources. Most of all, we learned that people wished to be informed. "Bring it on! with the anxiety, with the warnings, with everything,"(G5B) declared one of our participants from a post-ride focus group, expressing their excitement after being deeply moved by the CXR experience."
"Our conclusions suggest practical takeaways for further developing and deploying CXR experiences in these contexts and opportunities for future improvements. We compiled the participant’s suggestions into a draft community resiliency plan (added as an appendix). This draft will be disseminated among the participants and local authorities to initiate steps toward a formal community resiliency plan."
Given the nature of the tasks, maybe even include some literature on Collaborative VR and AR. Some relevant papers could be:
"Speicher, M., Hall, B. D., & Nebeling, M. (2019, May). What is mixed reality?. In Proceedings of the 2019 CHI conference on human factors in computing systems (pp. 1-15)."
"Marques, B., Silva, S., Alves, J., Araujo, T., Dias, P., & Santos, B. S. (2021). A conceptual model and taxonomy for collaborative augmented reality. IEEE transactions on visualization and computer graphics, 28(12), 5113-5133."
"Marques, B., Silva, S., Dias, P., & Santos, B. S. (2022). One-to-many remote scenarios: The next step in collaborative extended reality (XR) research. In Workshop on Analytics, Learning & Collaboration in eXtended Reality (XR-WALC). ACM International Conference on Interactive Media Experiences (IMX 2022) (pp. 1-6)."
"Sereno, M., Wang, X., Besançon, L., Mcguffin, M. J., & Isenberg, T. (2020). Collaborative work in augmented reality: A survey. IEEE Transactions on Visualization and Computer Graphics, 28(6), 2530-2549."
"Ens, B., Lanir, J., Tang, A., Bateman, S., Lee, G., Piumsomboon, T., & Billinghurst, M. (2019). Revisiting collaboration through mixed reality: The evolution of groupware. International Journal of Human-Computer Studies, 131, 81-98."
"Drey, T., Albus, P., der Kinderen, S., Milo, M., Segschneider, T., Chanzab, L., ... & Rukzio, E. (2022, April). Towards collaborative learning in virtual reality: A comparison of co-located symmetric and asymmetric pair-learning. In Proceedings of the 2022 CHI Conference on Human Factors in Computing Systems (pp. 1-19)."
"Han, E., Miller, M. R., Ram, N., Nowak, K. L., & Bailenson, J. N. (2022, May). Understanding group behavior in virtual reality: A large-scale, longitudinal study in the metaverse. In 72nd Annual International Communication Association Conference, Paris, France."
- Line ~135 - why mentioning AR in the middle of VR works? Separe both approaches? Given that the title is VR, maybe even remove AR?
- Line ~173 - DTs is used, but the acronym is never introduced.
- Line ~250 - a) or 1)? given that next, 2) and 3) are used.
- Lines 295 - 300 - there are ';' missing at the end of the bullets.
- Consider using italic from phrases quoted by the participants
- Most figures appear before being cited in the document. Please fix to appear only after being cited.
-- Table 1 is not cited in the manuscript
-- Figures 3 and 4 are not cited in the manuscript
-- Line 307 - Figure 7 is cited before Figure 6
-- Figure 7 - May be hard to understand by some readers. Consider enlarge letter size. Plus, mention the use of colour and possibly explain the meaning of each one.
-- Figure 10 is not cited in the manuscript.
- There are some mention of maintaining the anonymity through the manuscript. Given that authors names is already in the manuscript, I suggest removing this and cite what ever is necessary.
- Conclusions could benefit from some of the extra detailed given in the introduction
- Abbreviations - XR is not miXed Reality, but eXtended Reality
- Some references have URL, while most do not. I suggest removing all URL.
- Ref 56, 57 and 58 should be footnote.
- Avoid one last page with only 3 lines.
Author Response
Reviewer 2
Open Review
( ) I would not like to sign my review report
(x) I would like to sign my review report
Quality of English Language
( ) English very difficult to understand/incomprehensible
( ) Extensive editing of English language and style required
( ) Moderate English changes required
(x) English language and style are fine/minor spell check required
( ) I am not qualified to assess the quality of English in this paper
|
Yes |
Can be improved |
Must be improved |
Not applicable |
|
|
Does the introduction provide sufficient background and include all relevant references? We re-wrote the introduction, see highlighted parts on pages 1-2, |
( ) |
( ) |
(x) |
( ) |
|
Are all the cited references relevant to the research? |
(x) |
( ) |
( ) |
( ) |
|
Is the research design appropriate? |
(x) |
( ) |
( ) |
( ) |
|
Are the methods adequately described? |
(x) |
( ) |
( ) |
( ) |
|
Are the results clearly presented? |
(x) |
( ) |
( ) |
( ) |
|
Are the conclusions supported by the results? |
( ) |
(x) |
( ) |
( ) |
We re-wrote the conclusions and re-structured them according to the results structure logic.
Comments and Suggestions for Authors
Overall, the paper is very interesting, addressing a pertinent topic.It is well written and structured. Although it has 24 pages, I believe readers will have a good experience analysing it. The number of pages also reflect the amount of work that has been done. All in all, my only concern is associated with the introduction, besides that, the manuscript is quite mature.
Next, some comments and suggestions for improve the quality of the manuscript are described.
- With all due respect, I believe that is too much detail on the results being mentioned in the introduction. In contrast, there is not enough information regarding the topic being address. Hence, I urge that the author rephrase the introduction, providing more detail on the importance of the topic at hands, and the use of VR for such context, and reduce the amount of detail regarding the details. (Some of these details may be re-used for the conclusion for example, which is rather small).
-- I would go as far as to say that a similar approach could be applied to the abstract.
We thank reviewer 2 for their comments, and to follow their suggestion, we re-wrote the introduction, see highlighted parts on pages 1-2.
I suggest removing the following from the introduction:
"Our effort culminated in three sets of data: videos and transcripts of the focus groups, videos of the participants during the ride, and results from 74 pre- and post-ride surveys. We thematically analyzed the data for recurring themes.
In this paper, we show that although confronting face-to-face with the forecasted future evoked passing feelings of hopelessness, it was effective in improving community awareness of climate change: by bringing stakeholders together, providing information, and enhancing ownership and accountability. We argue that the immersive, embodied aspects of CXR make climate change feel real, its situational qualities bring climate change closer to home, and the communal aspects turn the spotlight to the community resources. Most of all, we learned that people wished to be informed. "Bring it on! with the anxiety, with the warnings, with everything,"(G5B) declared one of our participants from a post-ride focus group, expressing their excitement after being deeply moved by the CXR experience."
"Our conclusions suggest practical takeaways for further developing and deploying CXR experiences in these contexts and opportunities for future improvements. We compiled the participant’s suggestions into a draft community resiliency plan (added as an appendix). This draft will be disseminated among the participants and local authorities to initiate steps toward a formal community resiliency plan."
We thank reviewer 2 for this idea to make our introduction more concise and remove those parts from the introduction.
Given the nature of the tasks, maybe even include some literature on Collaborative VR and AR. Some relevant papers could be:
"Speicher, M., Hall, B. D., & Nebeling, M. (2019, May). What is mixed reality?. In Proceedings of the 2019 CHI conference on human factors in computing systems (pp. 1-15)." We added this source in the text and in the reference list.
"Marques, B., Silva, S., Alves, J., Araujo, T., Dias, P., & Santos, B. S. (2021). A conceptual model and taxonomy for collaborative augmented reality. IEEE transactions on visualization and computer graphics, 28(12), 5113-5133." We added this source in text and in the reference list.
"Marques, B., Silva, S., Dias, P., & Santos, B. S. (2022). One-to-many remote scenarios: The next step in collaborative extended reality (XR) research. In Workshop on Analytics, Learning & Collaboration in eXtended Reality (XR-WALC). ACM International Conference on Interactive Media Experiences (IMX 2022) (pp. 1-6)." We added this source in text and in the reference list.
"Sereno, M., Wang, X., Besançon, L., Mcguffin, M. J., & Isenberg, T. (2020). Collaborative work in augmented reality: A survey. IEEE Transactions on Visualization and Computer Graphics, 28(6), 2530-2549." We added this source in text and in the reference list.
"Ens, B., Lanir, J., Tang, A., Bateman, S., Lee, G., Piumsomboon, T., & Billinghurst, M. (2019). Revisiting collaboration through mixed reality: The evolution of groupware. International Journal of Human-Computer Studies, 131, 81-98."We added this source in text and in the reference list.
"Drey, T., Albus, P., der Kinderen, S., Milo, M., Segschneider, T., Chanzab, L., ... & Rukzio, E. (2022, April). Towards collaborative learning in virtual reality: A comparison of co-located symmetric and asymmetric pair-learning. In Proceedings of the 2022 CHI Conference on Human Factors in Computing Systems (pp. 1-19)." We added this source in the text and in the reference list.
"Han, E., Miller, M. R., Ram, N., Nowak, K. L., & Bailenson, J. N. (2022, May). Understanding group behavior in virtual reality: A large-scale, longitudinal study in the metaverse. In 72nd Annual International Communication Association Conference, Paris, France."After carefully considering this reference, we decided not to add it.
- Line ~135 - why mention AR in the middle of VR works? Separe both approaches? Given that the title is VR, maybe even remove AR? After carefully considering this reference, we decided not to add it.
- Line ~173 - DTs is used, but the acronym is never introduced.
We moved the Digital Twin to the introduction and introduced the DT acronym there.
- Line ~250 - a) or 1)? given that next, 2) and 3) are used.
All items are listed as numerical.
- Lines 295 - 300 - there are ';' missing at the end of the bullets.
Missing ‘;’ added
- Consider using italic from phrases quoted by the participants
We changed the participant's quotes to italic.
- Most figures appear before being cited in the document. Please fix it to appear only after being cited.
-- Table 1 is not cited in the manuscript
We added a reference to table 1, in line 211
-- Figures 3 and 4 are not cited in the manuscript
We added a text for Figure 3, line 221, and Figure 4 added to line 237.
-- Line 307 - Figure 7 is cited before Figure 6
Moved the figure to the place of first citation
-- Figure 7 - May be hard to understand by some readers. Consider enlarge letter size. Plus, mention the use of colour and possibly explain the meaning of each one.
-- Figure 10 is not cited in the manuscript. We added a reference to Figure 10 in line 522
There are some mention of maintaining the anonymity through the manuscript. Given that authors names is already in the manuscript, I suggest removing this and cite what ever is necessary.
We un-anonymous all our references to our forthcoming manuscript.
- Conclusions could benefit from some of the extra detailed given in the introduction
We thank reviewer 2 for this suggestion. Following it we completely rewrote the conclusions to follow the structure and content of the results.
- Abbreviations - XR is not miXed Reality, but eXtended Reality Corrected
- Some references have URL, while most do not. I suggest removing all URL. We removed all URLs
- Ref 56, 57 and 58 should be footnote. Corrected
- Avoid one last page with only 3 lines. Corrected

Reviewer 3 Report
1. The introduction should be corrected.
2. The objective and contribution could be written more clearly in the introduction section.
3. The literature analysis presented in the paper can be improved. It was prepared briefly and does not take into account all the aspects relevant to the scope of research included in the article.
4. Literature research should take into account all aspects relevant to the research topic.
These are examples of articles that can be included in the analysis of the literature: DOI: 10.1145/3307334.3326097; DOI: 10.1016/j.procs.2020.09.024.
5. The literature review could show the contribution of the paper. However, it is not clear what the paper's main contribution is.
6. I propose to include articles from the MDPI publishing house in the literature analysis.
7. It is unacceptable to group the items of literature in such a large number, eg: [9-13]. Line 25.
8. It is unacceptable to group the items of literature in such a large number, eg: [9-13]. Line 582.
9. The conclusions were not supported by the research results.
Author Response
Reviewer 3
Open Review
(x) I would not like to sign my review report
( ) I would like to sign my review report
Quality of English Language
( ) English very difficult to understand/incomprehensible
( ) Extensive editing of English language and style required
(x) Moderate English changes required
( ) English language and style are fine/minor spell check required
( ) I am not qualified to assess the quality of English in this paper
|
Yes |
Can be improved |
Must be improved |
Not applicable |
|
|
Does the introduction provide sufficient background and include all relevant references? |
( ) |
( ) |
(x) |
( ) |
|
Are all the cited references relevant to the research? |
( ) |
( ) |
(x) |
( ) |
|
Is the research design appropriate? |
(x) |
( ) |
( ) |
( ) |
|
Are the methods adequately described? |
( ) |
(x) |
( ) |
( ) |
|
Are the results clearly presented? |
( ) |
(x) |
( ) |
( ) |
|
Are the conclusions supported by the results? |
( ) |
( ) |
(x) |
( ) |
Comments and Suggestions for Authors
1. The introduction should be corrected.
2. The objective and contribution could be written more clearly in the introduction section.
3. The literature analysis presented in the paper can be improved. It was prepared briefly and does not take into account all the aspects relevant to the scope of the research included in the article.
We carefully edited the lit review and improved it according to the reviewers suggestions.
4. Literature research should take into account all aspects relevant to the research topic.
These are examples of articles that can be included in the analysis of the literature: DOI: 10.1145/3307334.3326097; DOI: 10.1016/j.procs.2020.09.024.
We carefully considered those references, and after adding Reviewer 2 and MDPI references, we found those additional references less relevant. We thank Revuwer 3 for this suggestion, and we will consider adding those references to our forthcoming paper.
5. The literature review could show the contribution of the paper. However, it is not clear what the paper's main contribution is. the contribution of the paper.
We especially thank Reviwe 3 for this comment, and we highlighted in the introduction that rewrote the introduction to clearly state what is paper 3 main contributions: considered those references, and after adding Reviewer 2 and MDPI references, we found those additional references are less relevant. We thank Revuwer 3 for this suggestion, and we will consider adding those references to our forthcoming paper.
6. I propose to include articles from the MDPI publishing house in the literature analysis.
We added the following MDPI citations to the paper:
%Calil, J., Fauville, G., Queiroz, A. C. M., Leo, K. L., Newton Mann, A. G., Wise-West, T., ... & Bailenson, J. N. (2021). Using virtual reality in sea level rise planning and community engagement—an overview. Water, 13(9), 1142.
% New MDPI citations we added:
%Scurati, G. W., Bertoni, M., Graziosi, S., & Ferrise, F. (2021). Exploring the use of virtual reality to support environmentally sustainable behavior: A framework to design experiences. Sustainability, 13(2), 943.
%Mirauda, D., Capece, N., & Erra, U. (2020). Sustainable water management: Virtual reality training for open-channel flow monitoring. Sustainability, 12(3), 757.
7. It is unacceptable to group the items of literature in such a large number, eg: [9-13]. Line 25. Corrected
8. It is unacceptable to group the items of literature in such a large number, eg: [9-13]. Line 582. Corrected
9. The conclusions were not supported by the research results.
We thank Reviewer 3 for this suggestion, and we completely rewrote the conclusions to follow the structure and content of the results.

Round 2
Reviewer 3 Report
The manuscript is revised according to my comments.
Author Response
Thank you.